# The Comparative Invasiveness of Endometriotic Cell Lines to Breast and Endometrial Cancer Cell Lines

**DOI:** 10.3390/biom13061003

**Published:** 2023-06-17

**Authors:** Katherine Ellis, Rachael Wood

**Affiliations:** 1Department of Chemical and Process Engineering, University of Canterbury, Christchurch 8041, New Zealand; rachael.wood@canterbury.ac.nz; 2Endometriosis New Zealand, Christchurch 8041, New Zealand; 3The Biomolecular Interaction Centre, University of Canterbury, Christchurch 8041, New Zealand

**Keywords:** endometriosis, peritoneal, ovarian, endometrial cancer, invasiveness, spheroids, EEC12Z, transwell

## Abstract

Endometriosis is an invasive condition that affects 10% of women (and people assigned as female at birth) worldwide. The purpose of this study was to characterize the relative invasiveness of three available endometriotic cell lines (EEC12Z, iEc-ESCs, tHESCs) to cancer cell lines (MDA-MB-231, SW1353 and EM-E6/E7/TERT) and assess whether the relative invasiveness was consistent across different invasion assays. All cell lines were subjected to transwell, spheroid drop, and spheroid-gel invasion assays, and stained for vimentin, cytokeratin, E-Cadherin and N-Cadherin to assess changes in expression. In all assays, endometriotic cell lines showed comparable invasiveness to the cancer cell lines used in this study, with no significant differences in invasiveness identified. EEC12Z cells that had invaded within the assay periods showed declines in E-Cadherin expression compared to cells that had not invaded within the assay period, without significant changes in N-Cadherin expression, which may support the hypothesis that an epithelial-to-mesenchymal transition is an influence on the invasiveness shown by this peritoneal endometriosis cell line.

## 1. Introduction

Endometriosis is a chronic condition [1] where cells similar to that of the endometrium implant in extra-uterine tissues [2,3]. It is a multifactorial disease of unknown etiology [4], with no present theory explaining all presentations of the disease [5]. Despite descriptions of the disease dating to 1500 BCE [6], endometriosis remains incurable [7]. Common treatment methods often fail to effectively relieve patient symptoms [8] and numerous knowledge gaps [9], such as the definitive cause of endometriosis, non-invasive means to diagnose the condition, and the origin of endometriosis pain remain.

Endometriosis is a non-neoplastic invasive condition [10], with the processes that allow endometriotic tissue to spread into the peritoneal cavity and beyond still requiring full elucidation [11]. One longstanding, predominant theory of endometriosis etiology is that of retrograde menstruation [12,13]. The retrograde menstruation theory posits that the endometrium is deposited from the uterus into the pelvic cavity via the fallopian tubes, where it becomes established as endometriotic lesions [14,15]. However, retrograde menstruation takes place in 90% of women (and people assigned female at birth) [16], while endometriosis impacts 6–18% of this population [17,18,19,20]. This theory also does not account for the rare presentation of endometriosis in men [21] and patients with Müllerian agenesis [22], including complete uterine agenesis [23], and similar disruptions to uterine development [24]. More recent theories include the metaplasia theory, that endometriosis originates from coelomic epithelium; the transplantation theory, that endometriotic cells have an invasive phenotype similar to metastatic tumours [25]; and the stem cell theory, that endometrial stem cells differentiate into endometriotic tissue [26]. The lack of a confirmed etiology for endometriosis means the factors that influence the establishment and spread of endometriosis are a vital area for ongoing research.

Endometriosis is believed to be able to spread via both lymphatic and hematogenous means to extra-pelvic locations as diverse as the umbilicus, thorax, vulva [27], brain, heart and liver [28]. Malignant transformations in extra-pelvic locations have been reported in both the gastrointestinal and urinary tracts [28]. Thoracic endometriosis involves the pleura or the lung parenchyma, primarily on the right side, with the most common presentation of the condition being pneumothorax from pleural lesions [28]. Pneumothorax from endometriosis can be cyclical in nature and align with menstruation (catamenial pneumothorax) [29]. Endometriosis is also associated with increased risks of ovarian [30,31,32], thyroid [32], breast and endometrial cancer [33] although the underlying relationships and causalities remain controversial and require further investigation. These aspects make an understanding of the mechanisms underlying endometriotic invasion vital in the development of future patient therapies.

While classified as a benign condition, endometriosis exhibits many of the hallmarks of cancer [34]. The hallmarks of cancer [35] shown by endometriosis include hyperproliferation [36,37], evasion of growth suppressors [38,39,40], apoptotic resistance [41,42], induction of angiogenesis [42,43,44], invasion and metastasis [25,27,28], and immunosurveillance evasion [40,45,46]. These features of the disease have resulted in a call to approach endometriosis not as a benign disease, but as an “atypical benign disease with invasive characteristics” [47]. 

A 1995 paper by Gaetje et al. (The Lancet) found that in collagen assays, cells from non-endometriotic endometria were non-invasive, while cells from peritoneal endometriotic lesions had similar invasive indices to the metastatic bladder carcinoma cell line EJ28 [25]. In another study, the invasion of stromal endometrial cells from endometriosis patients could be resisted by the peritoneal mesothelial cells of non-endometriosis patients (control) groups, but the stromal endometrial cells of both endometriosis patients and controls could invade the mesothelial peritoneal cells of endometriosis patients [48]. When endometriosis patient mesothelial peritoneal cells were invaded by endometrial stromal cells, the peritoneal cells lost their adhesion to one another, detached from the underlying substrate and underwent apoptosis when surrounded by endometrial stromal cells [48]. The invasion of endometriosis in extra-uterine tissues is a vital aspect of the pathology of the disease requiring further consideration and investigation.

One process thought to permit the invasion of endometriotic cells is the epithelial-to-mesenchymal transition (EMT). EMT involves reprogramming the epithelial cells which decreases adhesion, enhances invasion, and can be a contributing factor to the malignancy of epithelial tumours [49,50]. In endometriosis, there is an aberrant expression of cadherins, with increased expression of N-Cadherin, a path-finding cadherin [10], and decreased expression of E-Cadherin, an epithelial marker [51,52,53,54,55], a common pattern of EMT [49].

The purpose of this investigation is to characterize the invasiveness of peritoneal and ovarian endometriosis patient-derived cell lines and endometrial stromal cells from endometriosis patients, relative to the invasiveness shown by the triple-negative breast cancer cell line (MDA-MB-231), a hormone-driven endometrial cancer cell line (EM-E6/E7/TERT), and a bone chondrosarcoma cell line (SW1353). The intent of this avenue of analysis is to leverage the greater wealth of knowledge surrounding the invasion of cancers and to uncover more detail about the degree to which endometriosis is an invasive non-lethal metastatic disease.

## 2. Materials and Methods

### 2.1. Cell Culture

EEC12Z, an SV40-transformed epithelial endometriotic cell line derived from red peritoneal lesions [10], and iEc-ESCs, a stromal ovarian endometriosis cell line immortalized with human telomerase reverse transcriptase (hTERT) plasmids [56] were gifted by Asergally Fazleabas, PhD (Michigan State University, College of Human Medicine, Grand Rapids, MI, USA). tHESCs, a stromal endometrial cell line from endometriosis patients [57] was gifted by the Royal Women’s Hospital, Melbourne. MDA-MB-231, an invasive [58] epithelial mammary gland adenosarcoma cell line was gifted by the Mackenzie Cancer Research Group (Division of Pathology and Biomedical Science, University of Otago, Christchurch, New Zealand). EM-E6/E7/TERT, an endometrial cancer cell line immortalized by Kyo et al. in 2003 [59] was gifted by Logan Walker (Division of Pathology and Biomedical Science, University of Otago, Christchurch, New Zealand). SW1353, a bone chondrosarcoma cell line, was gifted by Callaghan Innovation (Christchurch, New Zealand). The negative control cell line was C2C12, a mouse myoblast cell line gifted by the Biological Application Technology Lab (Department of Electrical Engineering, University of Canterbury). All cell lines were maintained in Dulbecco’s modified Eagle medium (DMEM)/F12 (#11330057, Thermo Fisher Scientific, Waltham, MA, USA), supplemented with 10% fetal bovine serum [(FBS), #10091148, Thermo Fisher Scientific], 1% sodium pyruvate (#11360070, Thermo Fisher Scientific), and 1% penicillin/streptomycin, and incubated at 37 °C and 5% CO_2_.

#### 2.1.1. Cancer Cell Lines

MDA-MB-231 is a triple-negative cancer cell line that is one of the most studied breast cancer cell lines and it is known to be highly invasive [60,61,62]. This was selected for this work to represent the highest level of invasiveness. EM-E6/E7/TERT, the endometrial cancer cell line, was selected due to it being endometrial ‘like’ and, therefore, there may be some potential commonality with endometriosis [63]. Finally, SW1353 was chosen to ensure there was a moderately invasive cancer cell line that was unrelated to the anatomical area being studied [64].

#### 2.1.2. Endometriosis Cell Lines

EEC12Z was immortalized by Starzinski-Powitz et al. in Germany in 2001 [10]. Primary tissue samples were taken from patients undergoing laparoscopy. After tissue digestion, primary cells from biopsies of peritoneal bright red lesions were transfected with a SV40 plasmid which was introduced via electroporation. A total of 11 cell lines were developed that were determined to be epithelial-like (honey-comb morphology, expressing cytokeratins, vimentin and E-cadherin); however, only four were able to become true immortal cell lines. EEC12Z is the cell line that attained the highest passage number. The protein expression of EEC12Z was compared to primary tissue to confirm the behaviour was not significantly different after immortalization [10].

tHESCs were immortalized by Holdsworth-Carson et al. in Australia in 2019 [57]. As with EEC12Z, the primary tissue was isolated from patients undergoing laparoscopy. Immortalization was carried out using a Lenti-hTERT-green fluorescent protein virus with ViralPlus Transduction Enhancer and polybrene. The cell line was compared against primary hESC cultures using protein expression and in vitro assays such as proliferation, decidualization, and inflammatory response assays [57].

iEc-ESCs were immortalized by Fazleabas et al. in Michigan, USA in 2020 [56]. This cell line was also immortalized using hTERT technology; however, the primary tissue was obtained from a human ovarian endometriotic cyst. Stromal cells were isolated from the cyst and immortalized with an hTERT lentiviral vector. Characterization was carried out using protein expression, decidualization and karyotyping [56].

### 2.2. Transwell Invasion Assay

Transwell invasion assays allow the mimicking of the invasion of metastasizing cells through the basement membrane and are the gold standard of characterizing invasive cell behaviour [58]. A total of 2.0 × 10^5^ cells (in DMEM/F12 medium with 0.1% FBS) were seeded onto 24-well plate inserts with 8 μm pores (#CLS3464-48EA, Merck, Rahway, NJ, USA) coated with 30 μL of Geltrex (#A1413202, Thermo Fisher Scientific). The inserts were placed into wells of a 24-well plate containing 600 μL of 10% FBS DMEM/F12 medium. After 48 h of incubation, the medium was removed from the lower and upper chambers. An amount of 600 μL of methanol from the fridge was added to the lower chamber and the transwells were placed into the wells for ten minutes to fix the invading cells. The non-invading cells were removed from the Geltrex by adding 80 μL of refrigerated Corning cell recovery solution (#354253, Corning Inc, Corning, NY, USA) to the upper chamber. After 15 min of incubation at room temperature and triturating to break up the Geltrex, the solution from each well was transferred to 1.5 mL Eppendorf tubes and centrifuged for five minutes at 1000× *g*. The supernatant was removed, and the cells were plated on a 96-well plate (#167008, Thermo Fisher Scientific). The cells were incubated overnight to attach, then fixed with methanol from the fridge for ten minutes. To conduct a cell count to assess the degree of invasion [65,66,67], the transwells were removed from the methanol and dried at RT for fifteen minutes, then submerged in 1:1000 concentration Hoechst solution in phosphate-buffered solution (PBS) for ten minutes. The cell count was conducted by counting the number of cells in selected fields at nine different areas of the well at 200× total magnification.

### 2.3. Spheroid Drop Invasion Assay

The spheroid drop invasion assay was conducted in accordance with previously published methods [68]. A total of 1.5 × 10^5^ cells were centrifuged at 1000× *g* in a 1.5 mL Eppendorf tube. The supernatant was removed, and the cell pellet resuspended in 30 μL of Geltrex (#A1413202, Thermo Fisher Scientific). An amount of 10 μL of the suspended Geltrex cell solution was seeded onto the well of a 48-well plate (#150687, Thermo Fisher Scientific). The well plate was placed upside down into the incubator to incubate for 20 min at 37 °C and 5% CO_2_. Once removed from the incubator, 2 mL of DMEM/F12 with 10% FBS medium was added to each well. The medium was changed every 48 h. The spheroids were imaged daily and the area of the Geltrex spheroid and invading cells were measured in ImageJ.

### 2.4. Spheroid Gel Invasion Assay

Scaffold-less spheroids were constructed by adding 6.0 × 10^4^ cells from mono-layer culture to three wells of ultra-low attachment round-bottom 96-well plates (#CLS7007-24EA, Sigma-Aldrich, MO, USA) with 100 μL of medium and cultured for seven days [69] with half medium changes every two days [70]. On the seventh day, the medium was removed, and 50 μL of Geltrex was added to each well to surround the spheroid. The well plate was placed in the incubator for 60 min at 37 °C and 5% CO_2_. On top of the Geltrex gel, 200 μL of DMEM/F12 medium was added. The spheroids were imaged on day zero to determine the initial spheroid area. The invasion was characterized by measuring the change in the spheroid area and the sprouting area in ImageJ [11].

### 2.5. Immunofluorescence

Imaging was conducted using the EVOS M5000 Imaging System (#AMF5000, Thermo Fisher Scientific) with five images taken of each well at 400× total magnification. For each antibody, the definition of the positive threshold was selected in ImageJ based on a cell that was identified to be strongly positive. The same threshold was used consistently to identify the percentage of positive cells identified in each figure. To quantify the Cadherin staining, bright spots were treated as non-specific. To set the threshold, an image was chosen that had staining clearly associated with the cytoplasm of the cell. The threshold was set to identify this cell as positive, and this threshold was applied to all images. Cells were only counted as positive for Cadherin if the stain could be associated with the cytoplasm of the cell.

Between each staining step, the cells were washed with PBS by filling the well with PBS, leaving for five minutes, then removing and discarding the PBS. The cells were washed three times in each wash step. Cells were washed in PBS, fixed with cold methanol for 10 min, and blocked with 5% goat serum (#16210064, Thermo Fisher Scientific) in PBS for 30 min at room temperature (RT). Invasive assays were characterized with vimentin (#MA5-16409, Thermo Fisher Scientific), cytokeratin (#C5992-100UL, Sigma-Aldrich), N-Cadherin (#C3865-100UL, Sigma-Aldrich) and E-Cadherin (#Ab40772, Abcam). Cells were incubated with primary antibodies at 1:200 dilution in PBS for 90 min at RT. Secondary goat anti-Ms 594 (#A11032, Thermo Fisher Scientific) and secondary goat anti-Rb 488 (#A11017, Thermo Fisher Scientific) were added at 1:200 dilution in PBS, along with Hoechst (#H1398, Thermo Fisher Scientific) at 1:1000 dilution, to the cells for 45 min. The cells that had invaded in the transwell assay were fixed to the membrane of the Boyden chamber. The membrane was gently removed with a scalpel and placed into a 48-well plate for imaging.

#### Immunocytochemistry of Spheroid Drops

Cells were washed in PBS, and fixed with cold 1:1 acetone/methanol solution from the fridge for 10 min at −20 °C, and blocked with 5% goat serum in PBS overnight at 4 °C. Cells were incubated with primary antibodies overnight at 1:200 dilution in PBS at 4 °C. Secondary goat anti-ms 594 and secondary goat anti-rb 488 at 1:200 dilution in PBS were added to cells for 2 h at RT. Hoechst at a 1:1000 dilution with 20% glycerol and 20% Prolong^TM^ Gold Antifade Mountant (#P36934, Thermo Fisher Scientific) was added to the cells before imaging.

### 2.6. Area Measurements

Spheroid drop assays, scaffold-less spheroids and spheroid gel invasion were imaged using the EVOS M5000 Imaging System. The entire spheroid was imaged by taking multiple images at 100× total magnification. The images were stitched together using Hugin (macOS Version 10.16). The lens type was set to rectilinear, and the horizontal field of view was set to 1°. The images were saved without compression and the freehand tool was used in ImageJ to draw around the spheroid and the area taken up by the cells. To compare the areas over time, the percentage increase in spheroid areas relative to day zero was calculated for each spheroid.

### 2.7. Statistical Analysis

All experiments were repeated in triplicate, with the cells cultured separately for at least one passage for all cell lines. Data are presented as mean ± standard error of the mean [71]. All tests were carried out with a confidence level of 95% (α = 0.05) on raw data only. Unless stated otherwise, the null hypothesis was that the means of all levels are equal. Statistical analysis of the data was completed in GraphPad (version 9). Prior to conducting one or two-way ANOVA tests, normality was assessed using Shapiro–Wilks tests. In all figures, (*) indicates *p* < 0.05, (**) indicates *p* < 0.01, and (***) indicates *p* < 0.001. The post hoc tests were carried out by Bonferroni correction (multiple comparisons) and the Mann–Whitney test (pairwise comparison).

## 3. Results

### 3.1. Transwell Invasion Assay

The comparison of the level of invasion in the transwell assay is shown in Figure 1, along with the associated significance testing. The characterization of protein expression are summarized in Figure 2 and Figure 3. When the non-invaded and invaded protein expression of vimentin and cytokeratin were compared, the only statistically significant change was a decline in cytokeratin expression by EM-E6/E7/TERT (Figure 2). N-Cadherin expression was consistent across all cell lines, while there were significant declines in the expressions of E-Cadherin by MDA-MB-231, EM-E6/E7/TERT and EEC12Z (Figure 3).

### 3.2. Spheroid Drop Assay

All cancer and endometriosis-related cell lines exhibited increased invasive areas in the six-day period of cell culture (cell images in Figure 4, quantification in Figure 5). Predominantly, invasion out of the Geltrex drop was identifiable in all of these cell lines by day 2, with greater divergence in degree of invasiveness visible by day 4. By day 4, two clusters of invasiveness were identifiable. Cluster one consisted of MDA-MB-231, EM-E6/E7/TERT and EEC12Z, and cluster two consisted of SW1353, iEc-ESCs and tHESCs, with cluster one showing higher overall invasiveness than cluster two between days 4 and 6 (Figure 6).

### 3.3. Spheroid Gel Invasion Assay

#### 3.3.1. Self-Assembly of Spheroids

Cell lines C2C12, EM-E6/E7/TERT, SW1353, EEC12Z, iEc-ESCs, and tHESCs formed spheroidal, defined clusters within 24 h of being added to the low attachment round-bottom 96-well plates. The cell line MDA-MB-231 consistently did not form spheroidal clusters until five days in culture, at which point the margins were poorly defined relative to the well-established spheroids of the other cell lines (Figure 7). Throughout the seven days of spheroidal growth and establishment, there was a pronounced shrinkage in spheroid size of all cell lines. The percentage shrinkage relative to day one was similar amongst C2C12, EM-E6/E7/TERT, SW1353, iEc-ESCs, and tHESCs (61.0% to 72.3%), with less pronounced shrinkage exhibited by EEC12Z (45.7% ± 2.3%).

After six days in culture, C2C12 and SW1353 formed consistently small spheroids (0.25 ± 0.03 mm^2^, and 0.24 ± 0.02 mm^2^), while MDA-MB-231 and EEC12Z formed larger spheroids (0.85 ± 0.06 mm^2^, and 0.86 ± 0.04 mm^2^), and the spheroids formed by iEc-ESCs and tHESCs were intermediary in size (0.46 ± 0.07 mm^2^, and 0.30 ± 0.05 mm^2^) (Figure 7). The spheroids formed by EM-E6/E7/TERT were the most variable in size, with spheroids ranging in size from 0.29 mm^2^ to 0.77 mm^2^, and an SEM of 0.16 mm^2^ (Figure 7B).

#### 3.3.2. Gel Invasion of Spheroids

Most cell lines showed a steady increase in invaded area (Figure 7), indicating that cells from all endometriosis and cancer cell lines managed to escape relatively quickly as there was no delay in area invasion rate. EM-E6/E7/TERT appears to show a drop-off in invasion rate increase at day 5; however, it still overlaps with the invasion rate of tHESCs. To identify if their invasion rates are different from each other, further repeats would be required.

Similarly to the spheroid droplet assay, two groups can be seen forming when examining the percentage increase over time (Figure 7D). The group with the fastest invading cells is made up of EM-E6/E7/TERT and tHESCs, with the remaining cell lines showing similar invasion rates. The final invaded area splits the cell lines into three groups (Figure 7E). The highest increase in invasive area is by EM-E6/E7/TERT and tHESCs; however, examining the final value sees the appearance of a middle group of MDA-MB-231 and EEC12Z with an increased area value sitting between the top group and the rest of the cell lines. Finally, the smallest increase in area is carried out by C2C12, SW1353, and iEc-ESCs.

### 3.4. Cancer and Endometriosis Invasion Comparison

#### 3.4.1. Cancer and Endometriosis Invasion Comparison Using Transwell Assays

The cell lines that showed the greatest invasion, as determined by the number of Hoechst-stained nuclei present on the bottom of the membrane of the transwells, were the peritoneal endometriosis cell line EEC12Z with 90% ± 2% of cells invaded and by the breast cancer cell line MDA-MB-231 with 89% ± 4% of cells invaded within the 48 h of culture. Prior assessments of invasion of EEC12Z through transwells has shown an invasion of over 80%, comparable to the invasiveness shown by the bladder carcinoma cell line EJ28 [10]. MDA-MB-231 has previously been shown to be the most aggressively invasive and migratory in a comparison of breast cancer invasiveness [68].

With a one-way ANOVA analysis, there were no statistically significant differences between the percentage of cells that invaded within 48 h of culture between the breast cancer cell line MDA-MB-231, the endometrial cancer cell line EM-E6/E7/TERT, the peritoneal endometriosis cell line EEC12Z, the ovarian endometriosis cell line iEc-ESCs, or the endometrium cell line derived from endometriosis patient samples tHESCs (Figure 8). Cancer cell lines MDA-MB-231, EM-E6/E7/TERT, and endometriosis-related cell lines EEC12Z, iEc-ESCs, and tHESCs showed significantly more invasiveness than the SW1353 cell line (50.3% ± 8%).

#### 3.4.2. Cancer and Endometriosis Invasion Comparison Using Spheroid Drop Assays

When the percentage increase in spheroid drop and cell invasion area was determined on the final day of cell culture was compared, the cell line showing the highest invasiveness was again the peritoneal endometriosis cell line EEC12Z, with 206.7% ± 24.6% percentage area increase, followed by EM-E6/E7/TERT with 204.0% ± 21.5% percentage area increase. As with the transwell assays, there are no statistically significant differences in invasiveness between cancer cell lines MDA-MB-231, EM-E6/E7/TERT, and endometriosis-related cell lines EEC12Z, iEc-ESCs, and tHESCs. Unlike the transwell assays, there were no significant differences in the invasiveness of SW1353, and the other cancer cell lines or endometriosis-related cell lines.

#### 3.4.3. Cancer and Endometriosis Invasion Comparison Using Spheroid Gel Invasion Assays

The percentage increase in spheroid area through the formation of invasive edges was determined on the final day of cell culture and compared. The cell line showing the highest invasiveness was EM-E6/E7/TERT and tHESCs (Figure 7E) after the addition of Geltrex (131.3% ± 44.1% and 191.0% ± 60.0%, respectively). SW1353 (6.3% ± 2.7%) and iEc-ESCs (16.3% ± 4.1%) behaved similarly to the negative control C2C12 (2.7% ± 2.7%), with MDA-MB-231 and EEC12Z’s invasive area sitting in between these two groups (39.0% ± 12.4% and 49.3% ± 8.4%, respectively). The only cell line that showed statistical significance in the increase in area was tHESCs, which had a statistically larger increase in invaded area than all cell lines except EM-E6/E7/TERT. EM-E6/E7/TERT and tHESCs also showed the fastest increase in area invaded (shown in Figure 7D), which would be expected from the final invasion area value.

#### 3.4.4. Overall Invasive Behaviour Comparison

When comparing the pattern of invasiveness across all three assays (Figure 8), it is identifiable that the EEC12Z endometriosis cells showed comparably invasive potential to invasive cancer cell lines MDA-MB-231 and EM-E6/E7/TERT. The other two endometriosis-derived cell lines (iEc-ESCs, and tHESCs) were as invasive as MDA-MB-231 and EM-E6/E7/TERT using the gold standard of transwells, were similar to the bone chondrosarcoma cell line SW1353 when tracking invasion using spheroid droplets, and in the self-assembly assay, showed more invasive behaviour than SW1353. The overall pattern for endometriosis invasion showed these three endometriosis cell lines have a similar invasive potential to cancer cells.

## 4. Discussion

A previous study compared primary endometriosis cells to a metastatic bladder carcinoma cell line (EJ28) and assessed their invasive potential [25] by characterizing their invasion through collagen. This study has taken this concept further and examined key endometriosis cell lines by using gold standard invasive assays to characterize their behaviour compared to cancer cell lines with known invasiveness.

It has been postulated that the success of endometriosis cell invasion is related to the epithelial-to-mesenchymal transition [49,50]. This transition is often identified by the increase in N-Cadherin expression and decrease in E-Cadherin [49]. In this work, the invasive phenotype of all invaded cells (including cancer cell lines which are known to carry out this transition [72]), showed N-Cadherin expression was maintained; however, a decrease in E-Cadherin was observed. A decrease in E-Cadherin in cells has been shown to result in increased tumorigenicity and metastasis [73], as well as decreased rates of patient survival with cancer [74]. Despite no change in E-Cadherin expression, there were some significant changes observed in vimentin and cytokeratin, the former of which is a mesenchymal marker [72], the latter epithelial [75]. This change in E-Cadherin was only observed in epithelial-derived endometriosis cell lines. EMT is a stronger pattern than the mesenchymal-to-epithelial transition in relation to invasiveness [50] and therefore there is some evidence from this work that it may be occurring. Due to the short nature of the transwell experiment (48 h), it could be the case that there was insufficient time for the cells to undergo a full epithelial-to-mesenchymal transition. Future research should carry out longer invasive studies and undertake a full assessment of E-Cadherin and N-Cadherin expression. The current study was limited by access to relevant equipment.

The spheroid drop assay showed EEC12Z was behaving with similar invasiveness to two of the cancer cell lines (MDA-MB-231 and EM-E6/E7/TERT), and more invasively than the bone chondrosarcoma cell line SW1353. The extra-cellular matrix used here was Geltrex which is made up of laminin, collagen IV, entactin, and heparin sulfate proteoglycans [76]. This extracellular matrix (ECMs) is what the cells rely on for physical support and cell signalling; however, the remodelling activity carried out by cells can result in its degradation [77]. This remodelling is often carried out by cells that lead to pathological conditions such as invasive cancer as the breakdown of the ECM causes tissue destruction [78]. Additionally, the ECM can control cell migration and therefore any disruption can lead to the migration of inappropriate cell types [79]. Geltrex is made up of the required characteristics for cancer cell lines to grow (as it is derived from mouse tumours [76]); however, this information does not exist for endometriosis cells and therefore it may be that the invasive behaviour changes depending on the ECM and endometriosis cells could be more or less invasive if placed in different ECM conditions. A study in 2012 found that stromal endometrial cells from endometriosis patients could only invade endometriosis-derived tissue, and were unable to invade healthy tissue samples [48]. The make-up of the ECM environment is known to impact cell function, particularly the invasive behaviour of cancer [80,81], and further investigation into the in vivo environment of endometriosis cells and how it affects their invasive mechanism will be valuable information for building the etiology of this disease.

Recent studies have focused on organ-derived ECM using various decellularization methods to create materials to make scaffolds for recellularization [82]. These scaffolds have been used to create hydrogels and bioinks that have been showing success which is credited to the ECM gels possessing growth and differentiation factors that improve and regulate the cellular functions of the organ-related cells [83]. The generation of ECMs derived from the endometrium of both endometriosis and non-endometriosis patients, with a fully characterized storage and loss modulus, viscosity, and compressive modulus, could create in vitro models that are more reflective of the in vivo environment and thus add further value to this research.

The self-assembling spheroid assay successfully showed the ability not only for the invasion of the cells, but this is also the first time we are aware that two of the three cell lines (iEc-ESCs and tHESCs) have been shown to self-assemble into spheroids. A study in 2011 used the hanging drop method to create EEC12Z collagen I and Matrigel spheroids [11]. EEC12Z, in particular, has been shown here to form regular-sized spheroids and this method can be used to carry out 3D-focused cell behaviour. This behaviour is relevant to the classification of endometriosis, as most cancer cells lines readily form spheroids (as shown here); in fact, their inability to form spheroids is normally due to the loss of cell-cell adhesion molecules [84]. In addition, the spheroids allow for a better representation of the in vivo environment [85]. This ability to mimic the 3D environment allows for a better understanding of cell behaviour, cell morphology, drug sensitivity, and the aerobic impact on this behaviour as the cells on the inner part of the spheroid are subject to lower oxygen tensions than those on the outside [86]. Spheroids are currently used for testing potential drugs for cancer treatment, and it has been shown here that endometriosis cells could be examined in the same way.

Further work could be carried out to examine how the size of the cell influences the size of the spheroid formed. EEC12Z cells are one of the smallest cell lines in this experiment; however, they formed the largest spheroids. This could result in the EEC12Z cells requiring to migrate further through the Geltrex before they are able to proliferate outside of the spheroid. A similar pattern is seen with the MDA-MB-231 cells which could indicate their invasion rate was underestimated because the initial area is larger. Future work could examine tagging the cells, along with fluorescent cell cycle indicators, to track individual cell behaviour to better understand the mobility associated with each cell line.

Figure 7B highlights another area for improvement in this work in regard to the initial spheroid size. Although most cell lines were able to form consistently shaped spheroids, EM-E6/E7/TERT cells formed spheroids of more variable sizes. This is unusual as spheroids seeded with the same amount of cells are expected to form consistently sized spheroids (within the same cell line) [87]. In addition, when the invasion rate was compared to each spheroid, it was identified that the larger the EM-E6/E7/TERT spheroid, the lower the invasiveness. Further analysis (including functional and genetic mapping) should be carried out on this cell line to better understand its use as a cancer control for this work, as well as adding value to endometrial cancer research.

The future direction for this work includes the development of primary endometriosis spheroids and a comparison of the invasive behaviour of primary cells isolated from the different sub-types of endometriosis, as well as at different stages of disease. The stage of disease of endometriosis is classified in the same way as cancer (Stage I–IV) and it would be beneficial to identify how these factors impact the invasive behaviour of endometriosis cells. If endometriosis invasion can be compared more directly to cancer and similar invasive mechanisms can be identified, it is likely this will result in increased research funding for endometriosis and better awareness for this chronic condition that affects 6–18% of women (and those assigned female at birth). In addition, it could result in the translation of targeted cancer treatments to develop a cure for this disease.

## Figures and Tables

**Figure 1 biomolecules-13-01003-f001:**
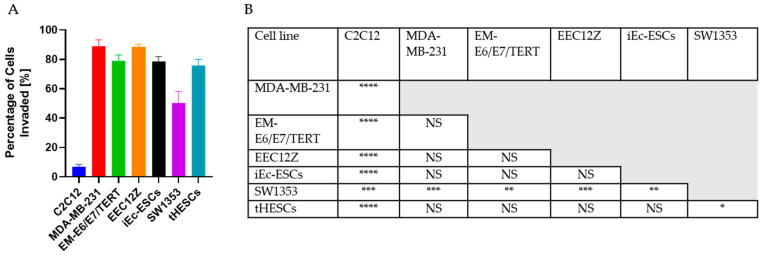
Analysis of the transwell assay (**A**) showed that the C2C12 cells were significantly (**B**) less invasive than all other conditions. Two endometriosis cell lines (EEC12Z, iEc-ESCs) and the endometrial cancer cell line EM-E6/E7/TERT) showed no significant difference from the triple invasive breast cancer cell line MDA-MB-231, and were significantly more invasive than the bone cancer cell line SW1353. Statistical significance is depicted by * *p* < 0.05, ** *p* < 0.01, *** *p* < 0.001, **** *p* < 0.0001 and is derived from a one-way ANOVA analysis and Bonferroni post hoc test.

**Figure 2 biomolecules-13-01003-f002:**
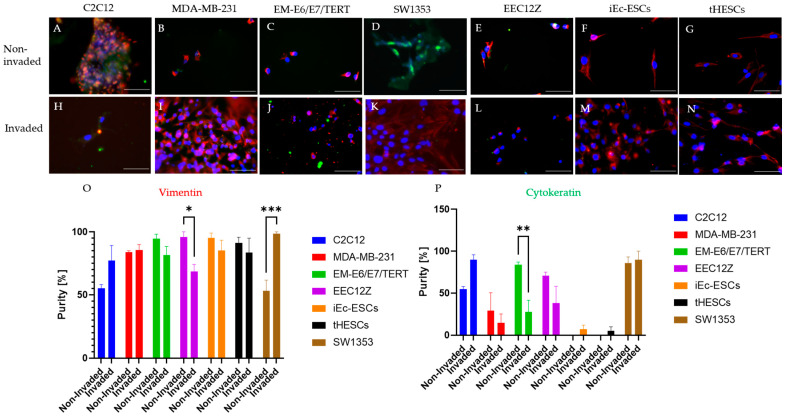
Immunofluorescence staining of transwell invasion assay for vimentin (red) and cytokeratin (green) of mouse myoblasts C2C12, breast cancer cells MDA-MB-231, endometrial cancer cells EM-E6/E7/TERT, peritoneal endometriosis cells EEC12Z, ovarian endometriosis cells iEc-ESCs, and endometrium from endometriosis patients tHESCs. Fluorescent images are shown for non-invaded cells across the top and invaded cells along the bottom including (**A**) non-invaded C2C12, (**B**) non-invaded MDA-MB-231, (**C**) non-invaded EM-E6/E7/TERT, (**D**) invaded SW1353, (**E**) non-invaded EEC12Z, (**F**) invaded iEc-ESCs, (**G**) non-invaded tHESCs, (**H**) invaded C2C12, (**I**) invaded MDA-MB-231, (**J**) invaded EM-E6/E7/TERT, (**K**) invaded SW1353, (**L**) invaded EEC12, (**M**) invaded iEc-ESCs, (**N**) invaded tHESCs, (**O**) quantification of Vimentin expression and (**P**) quantification of Cytokeratin expression. Statistical significance is depicted by * *p* < 0.05, ** *p* < 0.01, *** *p* < 0.001 and is derived from a two-tailed unpaired *t*-test. Scale bars represent 100 μm.

**Figure 3 biomolecules-13-01003-f003:**
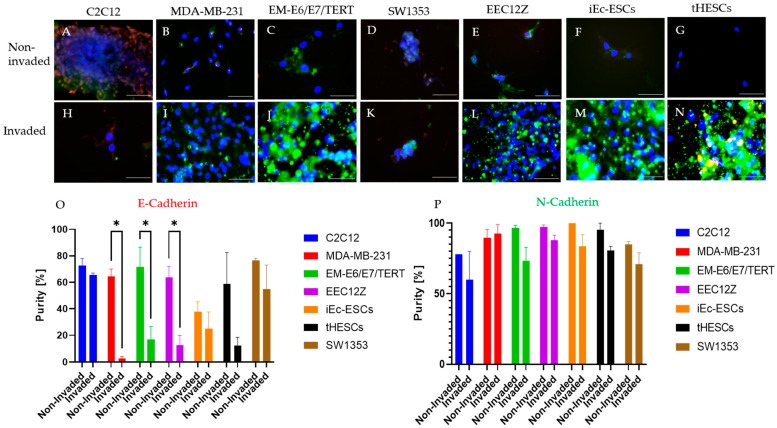
Immunofluorescence of N-Cadherin (green) and E-Cadherin (red) of transwell invasion assay of mouse myoblasts C2C12, breast cancer cells MDA-MB-231, endometrial cancer cells EM-E6/E7/TERT, peritoneal endometriosis cells EEC12Z, ovarian endometriosis cells iEc-ESCs, and endometrium from endometriosis patients tHESCs. Fluorescent images are shown for non-invaded cells across the top and invaded cells along the bottom including (**A**) non-invaded C2C12, (**B**) non-invaded MDA-MB-231, (**C**) non-invaded EM-E6/E7/TERT, (**D**) invaded SW1353, (**E**) non-invaded EEC12Z, (**F**) invaded iEc-ESCs, (**G**) non-invaded tHESCs, (**H**) invaded C2C12, (**I**) invaded MDA-MB-231, (**J**) invaded EM-E6/E7/TERT, (**K**) invaded SW1353, (**L**) invaded EEC12, (**M**) invaded iEc-ESCs, (**N**) invaded tHESCs, (**O**) quantification of E-Cadherin expression and (**P**) quantification of N-Cadherin expression. Statistical significance is depicted by * *p* < 0.05, and is derived from a two-tailed unpaired *t*-test. Scale bars represent 100 μm.

**Figure 4 biomolecules-13-01003-f004:**
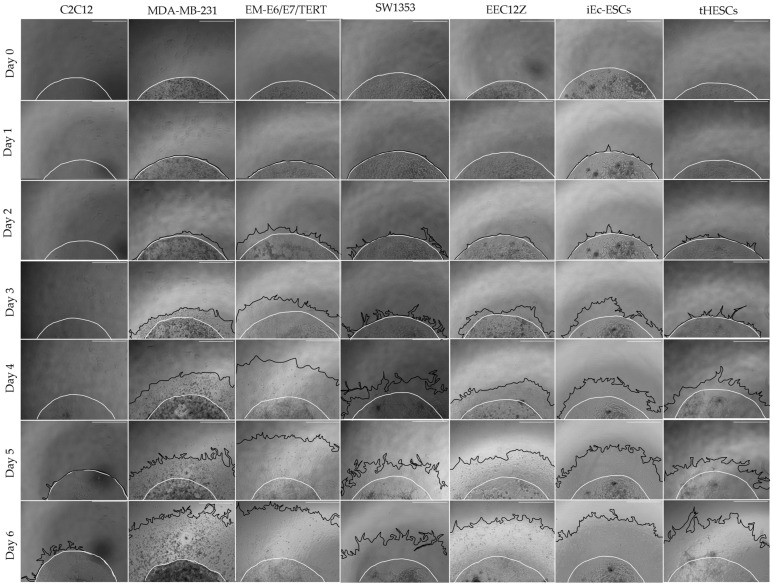
Spheroid drop images of mouse myoblasts C2C12, breast cancer cells MDA-MB-231, endometrial cancer cells EM-E6/E7/TERT, chondrosarcoma cells SW1353, peritoneal endometriosis cells EEC12Z, ovarian endometriosis cells iEc-ESCs, and endometrium cells from endometriosis patients tHESCs. A mixture of 5 × 10^4^ cells in Geltrex was seeded in triplicate as a drop on a 48-well plate for 6 days in culture with medium changes on days 2 and 4. The cells were imaged every day with the borders of the Geltrex shown in white and the borders of the invasive cells shown with black lines. Scale bars represent 1 mm.

**Figure 5 biomolecules-13-01003-f005:**
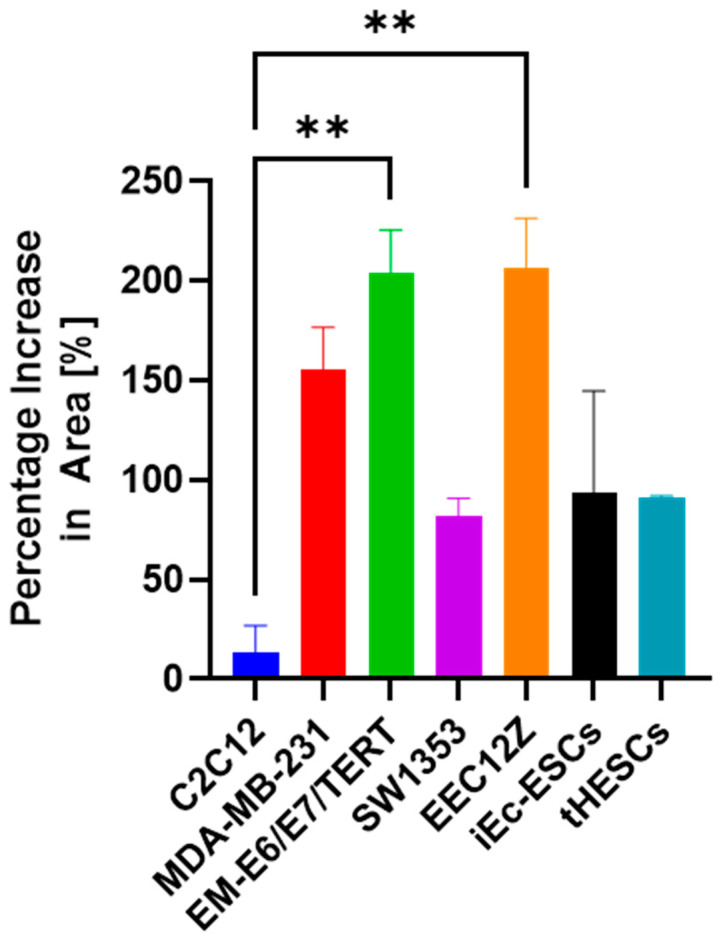
Analysis of the spheroid drop assay showed the MDA-MB-231 and EEC12Z cells were significantly more invasive than the C2C12 cells. Statistical significance is depicted by ** *p* < 0.01, and is derived from a one-way ANOVA analysis and Bonferroni post hoc test.

**Figure 6 biomolecules-13-01003-f006:**
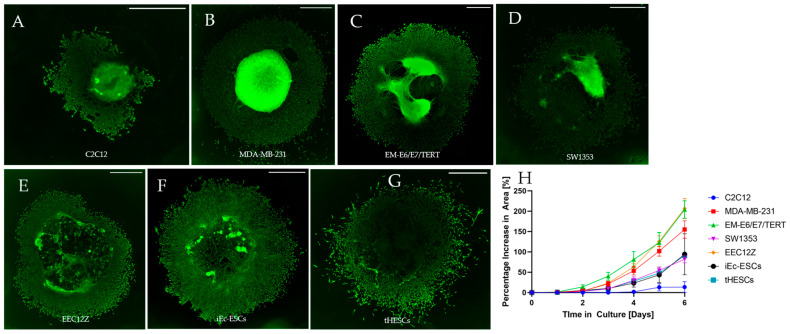
Immunofluorescence of spheroid drop assays stained for beta-actin (**A**) C2C12, and vimentin for (**B**) MDA-MB-231, (**C**) EM-E6/E7/TERT, (**D**) SW1353, (**E**) EEC12Z, (**F**) iEc-ESCs, and (**G**) tHESCs. The percentage increase in the area of spheroid drops for all cell lines over six days in culture is graphed and shown in (**H**). Scale bars represent 1 mm.

**Figure 7 biomolecules-13-01003-f007:**
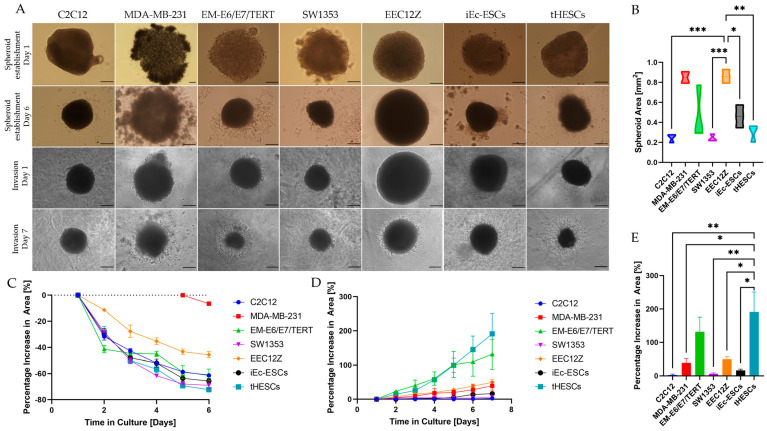
The spheroids self-assembled over six days of culture in a low attachment plate (**A**) and this is shown as the spheroid decreases in size over time as they form (**C**) to make spheroids of varying sizes between cell lines (**B**) with the MDA-MB-231 and EEC12Zs forming larger initial spheroids than C2C12, SW1353 and tHESCs. EM-E6/E7/TERT can be seen to have the most variety in initial spheroid size. After six days, the spheroids were covered in Geltrex, and cultured for a further six days. The invasion of the cells out of the spheroid through the Geltrex was tracked (**A**) and the percentage increase in area where the cells established themselves was measured over time (**D**) and compared to the size of the spheroid at day 6 (**E**). Statistical significance is depicted by * *p* < 0.05, ** *p* < 0.01, *** *p* < 0.001 and is derived from a one-way ANOVA analysis and Bonferroni post hoc test. Scale bars represent 250 μm.

**Figure 8 biomolecules-13-01003-f008:**
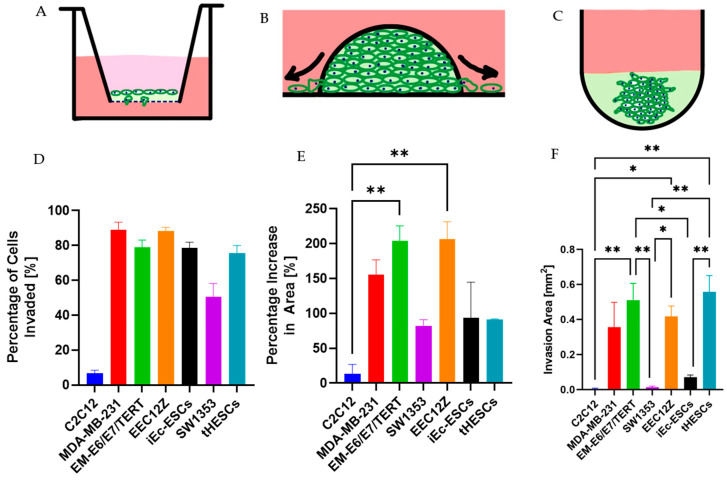
Cancer and endometriosis invasion comparison summary for the (**A**) transwell assay, (**B**) spheroid drop assay, and (**C**) spheroid gel invasion assay for C2C12, MDA-MB-231, EM-E6/E7/TERT, SW1353, EEC12Z, iEc-ESCs, and tHESCs. (**D**) Percentage of cells invaded through the transwell after 48 h of culture (significance table found in Figure 1B). (**E**) Percentage change in area on day six of culture for the spheroid drop assay. (**F**) Percentage change in area on day seven for the spheroid gel invasion assay. Error bars represent the standard error of the mean. * *p* < 0.05, ** *p* < 0.01, are derived from a one-way ANOVA analysis and Bonferroni post hoc test.

## Data Availability

The data presented in this study are available from the authors at reasonable request.

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
