# Peer review of "The Comparative Invasiveness of Endometriotic Cell Lines to Breast and Endometrial Cancer Cell Lines"

_biomolecules, 2023, doi:10.3390/biom13061003_

Round 1

Reviewer 1 Report

Nice work, in which the previously published results were verified at a new evidentiary level and using new methods. There are no comments on the formulation of the problem and the results obtained.  However, in my opinion, from the data obtained, a completely logical conclusion suggests itself that all epithelial cell lines (EEC12Z, MDA-MB-231, EM-E6/E7/TERT) form one cluster, and all stromal/mesenchymal (iEc- ESCs, tHESC) is different. With some restrictions, the C2C12 line can also be assigned to the second cluster. The discussion of EMT for mesenchymal lineages may not be entirely appropriate.

A few minor bugs have been noticed.

1.       1. In the caption to fig. 1 «endometriosis cell lines» are mentioned, which is not entirely correct, at least one of these lines is not directly related to endometriosis

2.       In the caption to fig. 1 «mouse fibroblasts NIH3T3» mentioned, not found anywhere else

3.       There is no reference in the text to Fig. 5

4.       3.4.1. It can hardly be argued that 90% ± 2 is ahead of 89% ± 4 J

The article can be published after making minor changes.

Author Response

Thank you for taking the time to review our manuscript. A summary of our changes is shown below.

However, in my opinion, from the data obtained, a completely logical conclusion suggests itself that all epithelial cell lines (EEC12Z, MDA-MB-231, EM-E6/E7/TERT) form one cluster, and all stromal/mesenchymal (iEc- ESCs, tHESC) is different. With some restrictions, the C2C12 line can also be assigned to the second cluster.

Thank you for this comment. It is definitely an interesting pattern, however, we believe that there is no definite pattern formed across all three assays to truly be able to draw a circle around these clusters. Also, we are still learning about endometriosis, we are not comfortable calling any of the endometrial cell lines purely epithelial simply based on their vimentin and N-Cadherin expression and are more confident describing them as ‘epithelial like’.

The discussion of EMT for mesenchymal lineages may not be entirely appropriate.

We have added a quantification for the relevance to epithelial lineages only (Line 41).

A few minor bugs have been noticed.

  1. In the caption to fig. 1 «endometriosis cell lines» are mentioned, which is not entirely correct, at least one of these lines is not directly related to endometriosis
  2. In the caption to fig. 1 «mouse fibroblasts NIH3T3» mentioned, not found anywhere else
  3. There is no reference in the text to Fig. 5
  4. 3.4.1. It can hardly be argued that 90% ± 2 is ahead of 89% ± 4 J

All these bugs have been addressed and corrected in the manuscript.

Reviewer 2 Report

Thank you for allowing me to review your work. This is not my area of expertise. However, your writing, presentation of the methods and results and discussion were well presented. I had no issues with the manuscript.

Author Response

Thank you for taking the time to review our manuscript. We appreciate your input.

There were no changes recommended.

Reviewer 3 Report

The manuscript by Ellis and Wood described their findings on the comparative similarities and difference between cells of a benign ‘invasive’ medical condition, endometriosis and certain cancer cells. Using some of the well established methods, the study has shown that the endometrial cells (immortalised) had similar invasiveness to some of chosen cancer cell lines and have further demonstrated that the both cell populations, namely endometrial cells and cancer cells, went through the EMT process during the invasion, in a similar patter, evidenced by certain EMT biomarkers including E- and N-cadherins. Overall, the study covers an very important medical condition, in that a benign condition exhibited ‘malignant like’ biological pattern, a phenomenon puzzled the medical and research communities for decades and yet without a solid conclusion. The study would contribute to this interesting subject area. There are, however, some important points to be clarified and addressed.

1.       Choice of endometrial cells. The study selected three endometrial cell lines, two of which were immortalised by different oncogenes. The study drew the conclusions from the comparison based on these three endometrial cells, which by the brief description sound rather different. A full description on the properties of these cells are necessary, including how they differ from the primary cells.

2.       Choice of cancer cells. The study chose one of well known breast cancer cell line and interestingly one endometrial cancer cell line as representatives of cancer cells. The rationale behind the selection needs careful explanation. In many mays, one could choose cancer cell lines with high-, low- or no- invasiveness, cell lines with similar and different tissue origins and with a good number.

3.       Methods and Results, Figure-2 and 3. Whilst some of the methods, particularly the three types of invasion assays, were given in rather nice details and easy to follow, others need additional information. For example, section 2.5.1 describes immunofluorescence staining methods. Not information was given on how the staining was quantified to support findings in Figures-3, which showed some quantitative staining of the cadherins. Cadherins have different distribution in the cell fractions, namely membranous vs cytoplasmic. Likewise, vimentin and cytokeratin are filamentous in the cytoplasmic. It was not sure how these filament staining was evaluated. It will be necessary to state the way that the fluorescence intensity was quantitated.

4.       Figure-6. The authors stained control cell, a mouse fibroblast cell for actin and stained the rest of the cells with vimentin. What the rationale to stain different cytoskeletal proteins in different cells. Fibroblasts are well known for their vimentin staining. Should A in this figure be a vimentin staining or at least dual image for both actin and vimentin?

5.       Figure-8. This is a rather nice and informative figure, except that statistical information is missing from D which did not indicate any statistical information and information was not available in the respective text either. Yet, some groups differs dramatically. It would be necessary to clarify this point and indicate the difference or if there deemed to be none.

Author Response

Thank you for taking the time to review our manuscript. We appreciate your input. A summary of the changes we have made is shown below.

  1. Choice of endometrial cells. The study selected three endometrial cell lines, two of which were immortalised by different oncogenes. The study drew the conclusions from the comparison based on these three endometrial cells, which by the brief description sound rather different. A full description on the properties of these cells are necessary, including how they differ from the primary cells.

This has been added in a new section “Section 2.1.2.”

  1. Choice of cancer cells. The study chose one of well known breast cancer cell line and interestingly one endometrial cancer cell line as representatives of cancer cells. The rationale behind the selection needs careful explanation. In many mays, one could choose cancer cell lines with high-, low- or no- invasiveness, cell lines with similar and different tissue origins and with a good number.

This has been added in a new section “Section 2.1.1.”

  1. Methods and Results, Figure-2 and 3. Whilst some of the methods, particularly the three types of invasion assays, were given in rather nice details and easy to follow, others need additional information. For example, section 2.5.1 describes immunofluorescence staining methods. Not information was given on how the staining was quantified to support findings in Figures-3, which showed some quantitative staining of the cadherins. Cadherins have different distribution in the cell fractions, namely membranous vs cytoplasmic. Likewise, vimentin and cytokeratin are filamentous in the cytoplasmic. It was not sure how these filament staining was evaluated. It will be necessary to state the way that the fluorescence intensity was quantitated.

Quantification for the staining is covered in the beginning of Section 2.5. Further detail for the Cadherin staining has been added in Lines 190-195.

  1. Figure-6. The authors stained control cell, a mouse fibroblast cell for actin and stained the rest of the cells with vimentin. What the rationale to stain different cytoskeletal proteins in different cells. Fibroblasts are well known for their vimentin staining. Should A in this figure be a vimentin staining or at least dual image for both actin and vimentin?

The staining is for mouse myoblasts not mouse fibroblasts (this was a typo in the original manuscript that has been corrected). We could not find any evidence of our myoblast line being positive for vimentin, so instead used actin to ensure we could see the cytoskeletons clearly in all conditions.

  1. Figure-8. This is a rather nice and informative figure, except that statistical information is missing from D which did not indicate any statistical information and information was not available in the respective text either. Yet, some groups differs dramatically. It would be necessary to clarify this point and indicate the difference or if there deemed to be none.

We have added into the figure legend that the significance can be found in Fig 1b. If preferred we can add the significance bars in or copy the table down. The reason it was chosen not to be in the graph is because the amount of significance bars makes the figure very messy and the significance becomes easily confused.